# NOISY MULTI-VIEW CONTRASTIVE LEARNING FRAMEWORK FOR ENHANCING TOP-K RECOMMENDATION

## ABSTRACT

Recommender systems have become an essential component of various online platforms, providing personalized recommendations to users. Collaborative filtering-based methods, such as matrix factorization, have been widely used to capture latent user-item preferences. Recently, graph-based methods have shown promising results by modeling the interactions between users and items as a graph and leveraging knowledge graphs (KG) to learn the user and item embeddings. Motivated by the recent success of contrastive learning in mining supervised signals from data itself, in this paper, we focus on establishing a noisy contrastive learning framework in Knowledge-aware recommendation systems and propose a self-supervised novel noisy multi-view contrastive learning framework for improving top-K recommendation. In this paper, we propose a novel recommendation system architecture that generates three different views of user-item interactions for improved recommendation along with a noise addition module. The global-level structural view leverages attention-based aggregation network Wang et al. (2019d) to capture collaborative information in the entity-item-user graph. In the item-item semantic view, we use a K-nearest Neighbour item-item semantic module to incorporate semantic relations among items. In the local view, we apply LightGCN He et al. (2020) with noisy perturbations to generate robust user-item representations. We then use two more signals such as representation loss and uniformity loss in positive pairs to improve the quality of the representations and ensure uniform representations in the representational space. Experimental results on two benchmark datasets demonstrate that our proposed method achieves superior performance compared to state-of-the-art methods. Additionally, we conducted extensive experiments on CTR task-based datasets to demonstrate the robustness of our framework's generalization in learning better user-item representations which can be seen in the supplementary material. All the codes to generate reproducible results are available in this anonymous repository[1]

## 1 INTRODUCTION

Recommender systems are instrumental in aiding users to unearth items of interest across various domains like e-commerce, social networking, and online advertising. Traditional approaches such as collaborative filtering (CF) Lian et al. (2018); He et al. (2017); Liu et al. (2014) mainly rely on historical user behavior data, like user-item interactions, to extract collaborative signals for recommendation. However, these methods often treat each interaction as an isolated instance, neglecting the inherent relations among them, leading to suboptimal performance in sparse data scenarios.

To mitigate these issues, integrating auxiliary information like knowledge graphs (KG) Wang et al. (2019d), which encapsulate rich factual data and connections about items, has been a prevalent strategy. This integration, known as knowledge-aware recommendation (KGR), facilitates enhanced user and item representations for recommendation. Despite substantial research efforts Wang et al. (2018b); Zhang et al. (2016b) in KGR, a core challenge remains: effectively harnessing the graph of item side (heterogeneous) information for latent user/item representation learning. Early endeavors Huang et al. (2018); Ai et al. (2018) primarily employed various knowledge graph embedding (KGE)

---

[1]https://anonymous.4open.science/r/NMCLK_Anonymous_Submission_Code

models to pre-train entity embeddings for item representation learning, but often faltered as they treated each item-entity relation independently, failing to distill adequate collaborative signals for item representations.

Motivated by these challenges, we propose the Noisy Multi-view Contrastive Learning Framework (NMCLK) to enhance Top-K recommendation. NMCLK aims to judiciously leverage limited user-item interactions and additional KG facts within a contrastive learning paradigm, which learns discriminative embeddings from unlabeled sample data by maximizing the distance between negative samples while minimizing the distance between positive samples. NMCLK, designed as an end-to-end knowledge-aware model, carefully balances the characteristics of both contrastive learning and knowledge-aware recommendation.

The NMCLK framework comprises three core modules: Multi-view generation, Contrastive module, and Alignment & Uniformity constraints. The Multi-view module exploits the rich collaborative and semantic information from Knowledge Graphs and user-item interactions, generating three complementary graph views. These views are supervised by a multilevel cross-view contrastive learning mechanism, aiming to enhance the consistency of embeddings across different views. Additionally, we integrate feature alignment and field uniformity constraints to further bolster the contrastive learning process.

Our contributions are encapsulated as follows:

- We introduce a model-agnostic contrastive learning framework – NMCLK, which can directly enhance the quality of feature representations in an end-to-end manner.

- Tailoring to the characteristics of the Top-K recommendation task, we design three self-supervised learning signals: contrastive loss, feature alignment constraint, and field uniformity constraint to augment contrastive learning performance.

- Through rigorous experimentation on two benchmark datasets, we demonstrate that our architecture achieves state-of-the-art performance, demonstrating its effectiveness

## 2 RELATED WORK

Recommendation systems offer personalized suggestions to users by analyzing their preferences and historical behaviors. Incorporating external knowledge sources, notably knowledge graphs (KGs), has emerged as a promising avenue to enhance user and item representations. This section delves into various knowledge-aware recommendation techniques, emphasizing embedding-based, path-based, and graph neural network (GNN)-based methods, and explores the burgeoning potential of contrastive learning in this realm.

**KG-Based Recommendation:** Embedding-based methods, such as CKE (Zhang et al., 2016a) and KTUP (Cao et al., 2019a), leverage knowledge graph embeddings (KGEs) to integrate learned entity and relation embeddings into the recommendation process. While effective, these methods sometimes prioritize semantic relatedness over recommendation accuracy. Path-based techniques, like PER (Yu et al., 2014) and KPRN (Wang et al., 2018c), harness KG connectivity patterns for recommendation. However, they often necessitate manually designed meta-paths, demanding domain expertise, especially for intricate KGs. GNN-based approaches, including KGCN (Wang et al., 2019c) and KGAT (Wang et al., 2019d), aggregate multi-hop neighboring nodes to capture graph structures and node features. Despite their efficacy, they predominantly employ supervised learning, relying on sparse original interactions. In contrast, our method harnesses self-supervised learning, capitalizing on inherent data signals to refine node representations.

**Contrastive Learning in Recommendation:** Contrastive learning has recently gained traction for enhancing user-item interaction representations. Models like HeCo (Wang et al., 2021c) and SGL (Wu et al., 2021) employ this paradigm. However, its integration with KG-aware recommendation systems remains nascent. Self-supervised contrastive learning, by leveraging data's intrinsic structure, offers potential for robust and generalizable KG representations, enhancing recommendation outcomes. Given KGs' semantic richness, pioneering contrastive learning techniques for KG-aware recommendations can unearth profound entity relationships, elevating recommendation system precision and efficiency.

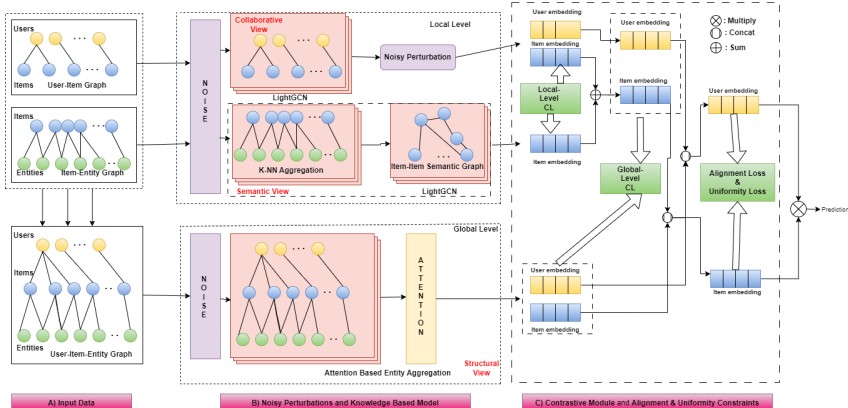

Figure 1: Architecture of the NMCLK framework and proposed model. NMCLK including two components: (a) noisy perturbations and a proposed model (which can be changed with another KG-based model); (b) a contrastive module and alignment & uniformity constraints.

## 3 TASK FORMULATION

In this section, we present the task formulation of our proposed Noisy Multi-view Contrastive Learning Framework for Enhancing Top-K Recommendation. We begin by introducing the necessary structural data, including user-item interactions and a knowledge graph.

**Interaction Matrix:** Let $U = u_1, u_2, \ldots, u_M$ and $V = v_1, v_2, \ldots, v_N$ be the sets of $M$ users and $N$ items, respectively, in a typical recommendation scenario. We define the user-item interaction matrix $Y \in \mathbb{R}^{M \times N}$ based on users' implicit feedbacks like any form of interaction but not explicit ratings, where $y_{uv} = 1$ indicates that user $u$ engaged with item $v$, such as clicking or playing a video, and $y_{uv} = 0$ otherwise.

**Knowledge graph:** In addition to the interaction data, real-world facts associated with items, such as item attributes or external commonsense knowledge, are stored in a knowledge graph $G$. The knowledge graph is a heterogeneous graph, where $G = (h, r, t) \mid h, t \in E, r \in R$, and $h, r, t$ denote the head, relation, and tail of a knowledge triple, respectively. $E$ and $R$ refer to the sets of entities and relations in G. For example, the triplet (Iron Man, film.film.star, Robert Downey Jr.) means that Robert Downey Jr. is a star of the movie Iron Man. In many recommendation scenarios, an item $v \in V$ corresponds to one entity $e \in E$, so we establish a set of item-entity alignments $A = (v, e) \mid v \in V, e \in E$. With the alignments between items and KG entities, the KG can profile items and offer complementary information to the interaction data.

**Task Description:** Given a user-item interaction matrix $Y$ and a knowledge graph $G$, our task is to learn a function that can predict how likely a user would adopt an item.

## 4 THE NMCLK FRAMEWORK

In this section, we elaborate on the different modules of the NMCLK model, which leverages self-supervised learning to improve user/item representation learning. Our proposed model is composed of three main components. Firstly, the Multi Views Generation module generates three different graph views, including global-level structural view, local-level collaborative and semantic view. Secondly, the Contrastive Module performs local-level and global-level contrastive learning to learn comprehensive and discriminative node embeddings. Finally, the Alignment and Uniformity Constraint Module enforces feature alignment and field uniformity constraints to facilitate the contrastive learning process.

## 4.1 Noisy Multi-view Generation Module

Our approach's methodology strategically partitions the user-item-entity graph into two separate graphs: the user-item graph and the item-entity graph. This bifurcation stems from the diverse item-item relationships inherent in the graph. The user-item graph serves as a collaborative view, emphasizing the identification of collaborative relationships between items via item-user-item co-occurrences. Conversely, the item-entity graph functions as a semantic view, targeting the exploration of semantic similarities between items through item-entity-item co-occurrences.

Distinct relationships between item entities are addressed by interpreting these relations in the knowledge graph as unique edge types linking items. These relations are seamlessly integrated into our contrastive learning framework by adopting multiple perspectives on item-item relationships. For instance, related hrt pairs such as (Hugh Jackman, ActorOf, Logan) and (JamesM., DirectorOf, Logan) are projected into both local and global contrasting views. The objective is to spatially approximate similar pairs while distancing dissimilar ones. The comprehensive user-item-entity graph is perceived as a structural view, striving to retain the entirety of path information, inclusive of the user-item-entity long-range connectivity. Drawing inspiration from the work of wu2022noisytune on pretrained language models, we adopt a matrix-wise perturbation technique. This method introduces varied uniform noises to distinct parameter matrices, contingent on the standard deviations of the parameters across different modules of the View encoders.

### 4.1.1 Collaborative View (CV) Encoder:

The collaborative view encoder captures collaborative information between items by modeling long-range connectivity from user-item interactions. Inspired by previous collaborative filter (CF) based work, we adopt Light-GCN to recursively perform aggregation K times. Light-GCN contains simple message passing and aggregation mechanisms without feature transformation and non-linear activation, which is effective and computationally efficient. The aggregation proceeding in the Kth layer can be formulated as:

$$\mathbf{e}_u^{(k+1)} = \sum_{i \in \mathcal{N}_u} \frac{1}{\sqrt{|\mathcal{N}_u||\mathcal{N}i|}} \mathbf{e}_i^{(k)}; \quad \mathbf{e}_i^{(k+1)} = \sum_{u \in \mathcal{N}_i} \frac{1}{\sqrt{|\mathcal{N}_u||\mathcal{N}i|}} \mathbf{e}_u^{(k)} \tag{1}$$

where $\mathbf{e}_i^{(k)}$ and $\mathbf{e}_u^{(k)}$ represent embeddings of user $u$ and item $i$ at the $k$-th layer, $\mathcal{N}_u, \mathcal{N}_i$ represent neighbors of user $u$ and item $i$ respectively. Then we sum representations at different layers up as the local collaborative representations $z_i^c$ and $z_u^c$, as follows:

$$z_u^c = \mathbf{e}_u^{(0)} + \cdots + \mathbf{e}_u^{(K)}; \qquad z_i^c = \mathbf{e}_i^{(0)} + \cdots + \mathbf{e}_i^{(K)} \tag{2}$$

Inspired by SimGCL simgcl and Noisytune wu2022noisytune papers' addition of random noise to the embeddings and model parameters which leads to robust representation learning we also perform a similar addition of noise to the generated user and item embeddings.

### 4.1.2 Semantic View Encoder:

To capture the semantic relationship between items, we introduce a $k$-Nearest-Neighbor item-item semantic graph, $S$. Constructed using a relation-aware aggregation mechanism, this graph retains both neighbor entities and relation data. In $S$, each entry $S_{ij}$ signifies the semantic similarity of item i to item j. If $S_{ij} = 0$, the items are unlinked.

Item representations are iteratively learned $K'$ times from the knowledge graph $\mathcal{G}$ using our relation-aware aggregating mechanism, which embeds both neighboring entities and their relations. For each triplet $(i, r, v)$, a relational message $\mathbf{e}r \odot \mathbf{e}v^{(k)}$ is crafted to convey varied triplet meanings, achieved by modeling the relation $r$ using projection or rotation operators [32].

Upon encoding item affinities in the semantic graph $S$, a Light-GCN with $L$ aggregation operations refines item representations. The message passing and aggregation in the $l$-th layer is given by:

$$\mathbf{e}_i^{(l+1)} = \sum_{j \in \mathcal{N}(i)} \widetilde{S} \mathbf{e}_j^{(l)} \tag{3}$$

Here, $\mathcal{N}(i)$ represents neighboring items, $\widetilde{S}$ is the normalized graph adjacency matrix from Equation 4, and $\mathbf{e}i^{(l)}$ is the item $i$ representation at layer $l$. The initial item representation $\mathbf{e}j^{(0)}$ is its ID embedding vector, as Light-GCN directly captures item affinities. The local semantic representations $\mathbf{z}_i^S$ are obtained by summing item representations across layers:

$$z_i^s = \mathbf{e}_i^{(0)} + \cdots + \mathbf{e}_i^{(l)} \tag{4}$$

### 4.1.3 STRUCTURAL VIEW ENCODER:

The Structural View Encoder comprises the Neighborhood Aggregator and the Structural Attention Network. The former captures structural details from each entity's vicinity. Given entities $N_i^{(k-1)}$ and their embeddings $\mathbf{e}j^{(k-1)}$ $j \in N_i^{(k-1)}$ at layer $(k-1)$, the aggregator computes:

$$\mathbf{h}_i^k = \sigma \left( \frac{1}{\left| N_i^{(k-1)} \right|} \sum_{j \in N(i)^{(k-1)}} \mathbf{W}^{(k)} \mathbf{e}_j^{(k-1)} \right) \tag{5}$$

Here, $\sigma(\cdot)$ is the activation function, and $\mathbf{h}_i^k$ represents aggregated neighbors of entity $i$ at layer $k$.

The Structural Attention Network emphasizes structural features based on semantic relationships within the KG. It computes attention weights:

$$\alpha_{ij}^{(k)} = \frac{\exp \left( \text{LeakyReLU} \left( \mathbf{a}^{(k)\top} \left[ \mathbf{W}^{(k)} \mathbf{e}_i^{(k-1)} | \mathbf{W}^{(k)} \mathbf{e}j^{(k-1)} \right] \right) \right)}{\sum\limits_{k \in \mathcal{N}_i} \exp \left( \text{LeakyReLU} \left( \mathbf{a}^{(k)\top} \left[ \mathbf{W}^{(k)} \mathbf{e}_i^{(k-1)} | \mathbf{W}^{(k)} \mathbf{e}_k^{(k-1)} \right] \right) \right)} \tag{6}$$

The final entity representation combines aggregated neighborhood and attention-weighted representations:

$$\mathbf{e}_i^{(k)} = \text{ReLU} \left( \mathbf{W}^{(k)} \left[ \mathbf{h}_i^k, \frac{1}{|\mathcal{N}_i|} \sum_{j \in \mathcal{N}_i} \alpha ij^{(k)} \mathbf{e}_j^{(k-1)} \right] \right) \tag{7}$$

Global representations $zu^g$ and $zi^g$ are obtained by summing all layers:

$$\mathbf{z}_u^g = \mathbf{e}_u^{(0)} + \cdots + \mathbf{e}_u^{\left( L' \right)} \quad ; \quad \mathbf{z}_i^g = \mathbf{e}_i^{(0)} + \cdots + \mathbf{e}_i^{\left( L' \right)} \tag{8}$$

In essence, the Structural View Encoder integrates neighborhood structural and KG semantic information to enhance entity representation. The Neighborhood Aggregator consolidates neighbor embeddings, while the Structural Attention Network determines attention weights based on semantic KG relationships.

## 4.2 CONTRASTIVE MODULE

Inspired by the advancements of self-supervised learning (SSL) in computer vision and natural language processing, we introduce contrastive learning (CL) for Top-K recommendation tasks to refine feature representations. This module is bifurcated into Local Level and Global Level CL.

### 4.2.1 LOCAL LEVEL CONTRASTIVE LEARNING

Given item $i$ with collaborative and semantic view embeddings $z_i^c$ and $z_i^s$, we employ local-level cross-view contrastive learning to enhance these representations. These embeddings undergo a transformation via a single hidden layer MLP:

$$z_{i-}^c = W^{(2)} \sigma \left( W^{(1)} z_i^c + b^{(1)} \right) + b^{(2)} \qquad z_{i-}^s = W^{(2)} \sigma \left( W^{(1)} z_i^s + b^{(1)} \right) + b^{(2)} \tag{9}$$

where $W^{(\cdot)}$ and $b^{(\cdot)}$ are trainable parameters, and $\sigma$ is the ELU function. Positive samples are formed by contrasting embeddings from the collaborative view with their counterparts in the semantic view. This results in the local-level contrastive learning loss $L^{local}$.

### 4.2.2 GLOBAL LEVEL CONTRASTIVE LEARNING

In the global level, node representations from global and local views are transformed similarly to the local level:

$$\mathbf{z}_{i-}^g = W^{(2)}\sigma\left(W^{(1)}\mathbf{z}_i^g + b^{(1)}\right) + b^{(2)}; \quad \mathbf{z}_{i-}^l = W^{(2)}\sigma\left(W^{(1)}\left(\mathbf{z}_i^c + \mathbf{z}_i^s\right) + b^{(1)}\right) + b^{(2)} \tag{10}$$

Using a similar positive and negative sampling strategy as the local level, the contrastive loss $\mathcal{L}^{global}$ is derived. The final objective combines losses from both views:

$$\mathcal{L}^{\text{global}} = \frac{1}{2N}\sum_{i=1}^N \left(\mathcal{L}_i^g + \mathcal{L}_i^l\right) + \frac{1}{2M}\sum_{i=1}^M \left(\mathcal{L}_u^g + \mathcal{L}_u^l\right) \tag{11}$$

The Contrastive Module integrates both local and global views to optimize entity representations.

### 4.3 ALIGNMENT AND UNIFORMITY CONSTRAINT MODULE

We drew inspiration from the methodologies presented in (Tongzhou Wang, 2022) and (Fangye Wang, 2022), and we adopt the alignment and uniformity constraints, which have been effectively utilized in CV and NLP domains to derive discriminative representations. To optimize these constraints, we construct positive and negative sample pairs. In the context of Top-K prediction tasks, features within the same "field" correspond to positive sample pairs, while those from different fields relate to negative sample pairs. Specifically, a "field" is defined as a distinct attribute or category associated with an item, such as genre, director, or actor in datasets like MovieLens. Conversely, a "feature" represents a specific instance of user interaction with items.

The alignment constraint ensures that features from the same field are closely represented in the feature space, highlighting their inherent correlations. In contrast, the field uniformity constraint aims to reduce the similarity between features from disparate fields. By enforcing this constraint, we ensure a compact representation of features within the same field in a low-dimensional spaceresentation space, emphasizing their inherent similarity. Conversely, the field uniformity constraint is designed to diminish the similarity between features across different fields, ensuring a clear distinction between them. By implementing this constraint, we ensure that features of a common field are densely packed in a low-dimensional space, enhancing their distinguishability.

#### 4.3.1 FEATURE ALIGNMENT

We introduce the feature alignment constraint to minimize the distance between features from the same field. The loss function for feature alignment is defined as follows:

$$\mathcal{L}_a = \sum_{f=1}^F \sum_{\mathrm{e_i, e_j} \in \mathrm{E}_f} \|\mathrm{e_i} - \mathrm{e_j}\|_2^2 \tag{12}$$

where $\mathrm{e_i}$ and $\mathrm{e_j}$ are two features from the same field, and $\mathrm{E}_f$ is the subset features of field $f$.

#### 4.3.2 FIELD UNIFORMITY

The field uniformity constraint, on the other hand, aims to minimize the similarity between features belonging to different fields. The loss function for field uniformity is defined as follows:

$$\mathcal{L}_u = \sum_{\mathrm{e_i} \in \mathrm{E}_f} \sum_{1 \le f \le F} \sum_{\mathrm{e_j} \in (\mathbb{E} - \mathrm{E}_f)} \mathrm{sim}\left(\mathrm{e_i}, \mathrm{e_j}\right) \tag{13}$$

Here, sim is the cosine similarity between two features. $\mathbb{E} - \mathrm{E}_f$ contains all features except those from field $f$. In Top-K recommendation systems, low-frequency and high-frequency features have equal chances to be considered in both feature alignment and field uniformity constraints. This helps to alleviate the suboptimal representation issue for low-frequency features when the two constraints are introduced during training.

## 4.4 Model Prediction

By performing multi-layer aggregation in three distinct views and optimizing with multi-level cross-view contrastive learning, we generate several representations for both users and items. Specifically, we obtain $z_u^c$ and $z_u^g$ for user $u$, and $z_i^c$, $z_i^s$, and $z_i^g$ for item $i$. To obtain the ultimate user/item representations, we combine these representations through summation and concatenation, and then employ inner product to predict their matching score.

$$z_u^* = z_u^g | z_u^c; \qquad z_i^* = z_i^g | (z_i^c + z_i^s); \qquad \hat{\mathbf{y}}(u,i) = z_u^{*\top} z_i^* \tag{14}$$

## 4.5 Multi-task Learning

In order to integrate the recommendation task with the self-supervised task, a multi-task training approach is employed to optimize the entire model. For the KG-aware recommendation task, a pairwise BPR loss is utilized to reconstruct the users' historical data. This loss function encourages the predicted scores of a user's previously interacted items to be higher than the unobserved ones. The training dataset, denoted by $O$, is comprised of observed interactions, denoted by $O^+$, and their unobserved counterparts, denoted by $O^-$. The BPR loss function is defined as follows:

$$\mathcal{L}_{\text{BPR}} = \sum_{(u,i,j) \in O} -\ln \sigma \left( \hat{\mathbf{y}}_{ui} - \hat{\mathbf{y}}_{uj} \right) \tag{15}$$

where $O = \{(u,i,j) \mid (u,i) \in O^+, (u,j) \in O^- \}$ is the training dataset consisting of the observed interactions $O^+$ and unobserved counterparts $O^-$; $\sigma$ is the sigmoid function.

We now combine BPR loss, local contrastive loss, global contrastive loss, field uniformity, and feature alignment loss. The hyperparameters for our combined loss function were meticulously chosen to ensure balanced contributions from each individual loss term. Specifically, they were set to harmonize the magnitudes of the losses, preventing any single term from dominating. This balanced approach ensures robust and discriminative feature representations. For clarity, the hyperparameters were set as follows: $\alpha = 0.2$, $\beta = 0.1$, $\lambda = 0.01$, $\gamma = 0.5$, and $\delta = 0.05$. We minimize the following objective function to learn the model parameter:

$$\mathcal{L}_{NMCLK} = \mathcal{L}_{\text{BPR}} + \beta \left( \alpha \mathcal{L}^{\text{local}} + (1-\alpha)\mathcal{L}^{\text{global}} \right) + \gamma \left( \delta \mathcal{L}_a + (1-\delta)\mathcal{L}_u \right) + \lambda \|\Theta\|_2^2 \tag{16}$$

where $\Theta$ is the model parameter set, $\alpha$ is a hyper parameter to determine the local-global contrastive loss ratio, $\beta$, $\gamma$, $\delta$ and $\lambda$ are five hyper parameters to control the contrastive loss, feature alignment, and uniformity loss, feature alignment-uniformity ratio, and $L_2$ regularization term, respectively.

# 5 Experiments

## 5.1 Experimental Setup

### 5.1.1 Datasets:

We evaluate the effectiveness NMCLK by conducting experiments on two popular benchmark datasets: **ML-100K**[2] and **ML-1M**[3], which are publicly available and have been used in existing works (Lv et al., 2021; Wang et al., 2019d;a;c). Table 1 displays the statistics of the three datasets mentioned above, all of which utilize the 10-core approach to filter out low-frequency users and items that appear less than ten times. This filtering ensures the quality of interaction data. Similarly, following the approach of KGAT (Wang et al., 2019d), 80% of interactions are randomly assigned for training, while the remaining 20% are assigned for testing. Moreover, for each user, 10% of interactions from the training set are chosen randomly as a validation set to fine-tune the hyper-parameters. Detailed descriptions of datasets can be found out in the links given in the footnote.

---

[2]https://grouplens.org/datasets/movielens/100k/
[3]https://grouplens.org/datasets/movielens/1m/

| Dataset | | ML-100K | ML-1M |
|---|---|---|---|
| User-Item Interaction | #users | 944 | 6041 |
| | #items | 1599 | 3656 |
| | #interactions | 97554 | 997580 |
| Knowledge Graph | #entities | 34629 | 79348 |
| | #relations | 26 | 51 |
| | #triplets | 91631 | 385923 |

Table 1: Dataset Statistics

### 5.1.2 EVALUATION METRICS:

In our evaluation of top-K recommendation and preference ranking tasks, we utilize four widely recognized metrics: **NDCG@K, MRR@K, Hit@K**, and **Recall@K**, where K is set to **20**. For every user in the test set, NDCG@K is a normalized discounted cumulative gain metric that considers the position of correctly recommended items. MRR@K stands for mean reciprocal rank, which considers the rank position of the first relevant item. Specifically, it calculates the reciprocal rank of the first correct item, which is then averaged over all users in the testing set. In other words, it measures the quality of the first recommended item that matches the user's preference. Hit@K measures the percentage of users who have at least one item among the top K recommended items that match their ground truth preference. It is a binary metric, where 1 indicates the existence of at least one relevant item among the top K recommendations, and 0 otherwise. Recall@K denotes the proportion of their rated items that surface in the top K recommended items.

### 5.1.3 BASELINES:

To validate the efficacy of our proposed NMCLK model, we juxtaposed it against a spectrum of state-of-the-art recommender system techniques. These encompass: **BPR-MF**(Rendle et al., 2009), a CF-based method employing Bayesian Personalized Ranking (BPR) loss tailored for implicit feedback in collaborative filtering; **CKE**(Zhang et al., 2016a), an embedding technique that synergistically combines collaborative filtering with TransR-derived knowledge embeddings, thereby encapsulating diverse knowledge forms such as structural, textual, and visual; **RippleNet**(Wang et al., 2018a), an innovative embedding-based approach that leverages a memory-network strategy to propagate user preferences on the knowledge graph (KG); **MKR**(Wang et al., 2019b), a method that seamlessly integrates KG embedding with recommendations, adeptly addressing challenges like data sparsity and cold starts; **KTUP**(Cao et al., 2019b), a dual-purpose model that learns KG completion and recommendation in tandem, harnessing KG relations to gain deeper insights into user-item preferences; **KGCN**(Wang et al., 2019c), a GNN-based technique that aggregates pivotal KG information to refine item embeddings, with a focus on accentuating user preferences; **KGNN-LS**(Wang et al., 2019a), another GNN model with the added advantage of label smoothness regularization, ensuring enriched item embeddings; **KGAT**(Wang et al., 2019d), a GNN strategy that incorporates attention mechanisms, facilitating iterative integration of user-item-entity graph neighborhoods; and **KGIN**(Wang et al., 2021b), an avant-garde GNN approach that delves into user intention by leveraging auxiliary item knowledge through a user-intent-item-entity graph. Each of these models offers unique methodologies and perspectives, making the comparative analysis comprehensive and robust.

| Model | MovieLens-100k | | | | MovieLens-1M | | | |
|---|---|---|---|---|---|---|---|---|
| | NDCG@20 | Hit@20 | MRR@20 | Recall@20 | NDCG@20 | Hit@20 | MRR@20 | Recall@20 |
| BPRMF | 0.2421 | 0.8706 | 0.3874 | 0.3259 | 0.2047 | 0.822 | 0.3533 | 0.2196 |
| CKE | 0.264 | 0.8706 | 0.3861 | 0.3481 | 0.1987 | 0.825 | 0.3429 | 0.218 |
| RippleNet | 0.1837 | 0.7911 | 0.2576 | 0.2707 | 0.1216 | 0.7056 | 0.2081 | 0.1425 |
| MKR | 0.2334 | 0.8462 | 0.3451 | 0.309 | 0.1919 | 0.8156 | 0.3281 | 0.2107 |
| KTUP | 0.227 | 0.825 | 0.35 | 0.297 | 0.2055 | 0.8273 | 0.3582 | 0.2188 |
| KGCN | 0.2564 | 0.8653 | 0.3818 | 0.3393 | 0.1942 | 0.8126 | 0.3386 | 0.2099 |
| KGNN-LS | 0.2565 | 0.86 | 0.3805 | 0.3393 | 0.1942 | 0.8126 | 0.3379 | 0.2093 |
| KGAT | 0.2562 | 0.8696 | 0.407 | 0.3452 | 0.2035 | 0.8323 | 0.3531 | 0.2217 |
| KGIN | 0.2419 | 0.8653 | 0.3475 | 0.3248 | 0.2011 | 0.8296 | 0.3592 | 0.2214 |
| **NMCLK** | **0.279** | **0.878** | **0.4156** | **0.3528** | **0.2313** | **0.8598** | **0.3958** | **0.2484** |

Table 2: The results of baselines in Top-K prediction. The best results are in boldface and the second best results are underlined.

### 5.1.4 PARAMETER SETTINGS:

We kept the embedding dimension at 64, used Adam (Kingma & Ba, 2017) optimizer, and set the batch size to 2048 for our model. We initialized the model parameters randomly using the Xavier (Glorot & Bengio, 2010) initializer. To stabilize and accelerate model training, we used pre-trained MF embeddings, as suggested in previous studies. We followed the suggestions (Lv et al., 2021; Wang et al., 2019d; 2021b) from the original papers or conducted empirical research to set the hyperparameters for each baseline model. The experimental setting for each is kept the same as mentioned in their respective papers.

## 5.2 PERFORMANCE COMPARISON

**Superiority of NMCLK:** The NMCLK model consistently outperforms all other models across both datasets. Specifically, on the MovieLens-100k dataset, NMCLK improves the NDCG@20 by approximately 8.9% over the next best-performing model, KGAT. This superior performance can be attributed to its unique framework that seamlessly integrates multiple views and methodologies to capture both collaborative and semantic signals from the knowledge graph.

**The Way of Exploiting KG Information via Multiple Views:** The NMCLK model's ability to exploit knowledge graph (KG) information through multiple views, particularly the local and global contrastive views, sets it apart from other models. For instance, models like BPRMF and CKE, which achieve NDCG@20 scores of 0.2421 and 0.264 respectively on MovieLens-100k, do not incorporate such a multi-view approach, leading to their inferior performance.

**Distinguishing Collaborative and Semantic Signals:** NMCLK's strength lies in its capability to distinguish between collaborative and semantic signals in the KG. While models like KTUP and MKR might capture either collaborative or semantic signals, our approach ensures that both types of signals are harnessed effectively. This is evident in its higher performance metrics, where NMCLK's Recall@20 score of 0.3528 on MovieLens-100k surpasses that of KTUP's 0.297 and MKR's 0.309.

**Role of Feature Alignment and Uniformity:** On the MovieLens-1M dataset, NMCLK's NDCG@20 score of 0.2313 is approximately 11.6% higher than that of KGCN, highlighting the importance of these features. The alignment ensures that features from the same field are closely distributed, enhancing the model's ability to make accurate recommendations. In contrast, the uniformity constraint ensures diverse feature representations.

## 6 CONCLUSION AND FUTURE WORK

In this paper, we presented the Noisy Multi-view Contrastive Learning Framework (NMCLK) aimed at enhancing top-K recommendation performance. The framework utilizes a contrastive learning module to merge item representations from multiple views, including textual and visual aspects. We introduced a novel noise injection strategy to add noise to the model parameters and cross-view representations, thereby improving the robustness and generalizability of the obtained representations. Moreover, we proposed a representation uniformity loss and feature alignment constraint to further refine the quality and consistency of the learned representations.

Through rigorous experimentation on two benchmark datasets, we demonstrated the superior performance of NMCLK over state-of-the-art methods across different evaluation metrics. Despite its promising results, there are several areas for improvement. Future work could explore various noise injection strategies and assess their impact on model performance. While our emphasis was on enhancing embedding consistency across different views through the contrastive learning framework, investigating the balance between consistency and uniqueness in multi-view embeddings is warranted. The effect of employing different modalities and their combinations on NMCLK's performance also deserves exploration. Additionally, alternative contrastive learning approaches like InfoNCE or SwAV could be explored for better item representation learning. The integration of attention mechanisms to highlight the feature importance of different views is another promising avenue. Lastly, extending NMCLK to address more complex recommendation scenarios such as cross-domain, sequential, or group recommendations could lead to more powerful recommendation systems. This work sets a strong foundation for these explorations, and we believe that these directions will contribute to more robust and effective recommendation systems.

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
