# NOISY MULTI-VIEW CONTRASTIVE LEARNING FRAMEWORK FOR ENHANCING TOP-K RECOMMENDATION

## 1 SUPPLEMENTARY MATERIAL

### 1.1 COMPATIBILITY ANALYSIS:

To verify the compatibility of NMCLK, we deploy it into other models, such as BPR-MF, CKE, KGNN-LS, KGAT, and KGIN. The results are shown in Table 1.

Firstly, learning Knowledge graph representation with NMCLK can significantly improve the performance of Top-k recommendation. Applied with NMCLK, the performance of base models is remarkably boosted, which confirms our hypothesis of improving the performance of Top-K prediction models by improving the quality of the features representations and demonstrates the effectiveness of NMCLK as seen in Table 1.

Furthermore, the contrastive module in NMCLK, which incorporates both Local Level Contrastive Learning and Global Level Contrastive learning, plays a significant role in generating high-quality feature representations. By leveraging this module, NMCLK can learn to generate feature embeddings that capture meaningful relationships between items and users in the graph. Moreover, the alignment and uniformity constraint module is also crucial in regulating feature representations during training. By introducing the feature alignment constraint, NMCLK can minimize the distance between features from the same field and encourage closely distributed feature representations for similar items or users. Additionally, the field uniformity constraint can minimize the similarity between features belonging to different fields, encouraging diverse feature representations. Overall, the combination of these techniques in the NMCLK framework leads to significant improvements in Top-k recommendation performance compared to the base models.

| Model | MovieLens-100K | | | | MovieLens-1M | | | |
|---|---|---|---|---|---|---|---|---|
| | NDCG@20 | Hit@20 | MRR@20 | Recall@20 | NDCG@20 | Hit@20 | MRR@20 | Recall@20 |
| BPRMF | 0.2421 | 0.8706 | 0.3874 | 0.3259 | 0.2047 | 0.822 | 0.3533 | 0.2196 |
| $NMCLK_{BPRMF}$ | **0.2544** | **0.8738** | **0.3924** | **0.329** | **0.2112** | **0.8237** | **0.3673** | **0.2211** |
| CKE | 0.264 | 0.8706 | 0.3861 | 0.3481 | 0.1987 | 0.825 | 0.3429 | 0.218 |
| $NMCLK_{CKE}$ | **0.2704** | **0.878** | **0.4019** | 0.3454 | **0.225** | **0.8421** | **0.3951** | **0.2318** |
| KGNN-LS | 0.2565 | 0.86 | 0.3805 | 0.3393 | 0.1942 | 0.8126 | 0.3379 | 0.2093 |
| $NMCLK_{KGNNLS}$ | **0.257** | **0.8664** | **0.3849** | **0.3397** | **0.1961** | **0.8157** | **0.3455** | **0.2095** |
| KGAT | 0.2562 | 0.8696 | 0.407 | 0.3452 | 0.2035 | 0.8323 | 0.3531 | 0.2217 |
| $NMCLK_{KGAT}$ | **0.2696** | **0.8738** | **0.3908** | **0.3409** | **0.2277** | **0.8427** | **0.3947** | **0.2357** |
| KGIN | 0.2419 | 0.8653 | 0.3475 | 0.3248 | 0.2011 | 0.8296 | 0.3592 | 0.2214 |
| $NMCLK_{KGIN}$ | **0.2748** | **0.8834** | **0.3896** | **0.3511** | **0.2243** | **0.8392** | **0.4096** | **0.2326** |

Table 1: Compatibility study of NMCLK framework

### 1.2 COMPONENT ANALYSIS:

We performed a component analysis of the NMCLK model to evaluate the contribution of each individual component to the final performance. We evaluate the model on two datasets, MovieLens-100K and MovieLens-1M, and report the results in Table 2.

The NMCLK model, in its full configuration, incorporates both Local and Global Contrastive Learning (CL), along with Representation Uniformity features such as Alignment and Uniformity. When evaluated on the MovieLens-100k and MovieLens-1M datasets, this configuration showcases optimal performance, setting a benchmark for subsequent model variations.

Upon removing the noise from the NMCLK model, there's a discernible decrease in performance across both datasets. This suggests that noise plays a pivotal role in enhancing the model's robustness and generalization capabilities. The significance of Contrastive Learning is further underscored when both Local and Global CL are excluded, leading to a performance drop. However, the decline isn't as pronounced as one might expect, indicating that while CL is beneficial, other components of the model also contribute substantially to its efficacy. The importance of Representation Uniformity is highlighted when the Alignment feature is removed. The model's performance experiences a slight dip, emphasizing that aligning features from the same field is crucial for generating closely distributed feature representations for similar items or users.

Interestingly, when the Global view is excluded in various configurations, there's a consistent decline in performance across all metrics and datasets. This underscores the Global view's critical role in capturing broader contextual information, which is instrumental in enhancing recommendation quality. In configurations where only the Local view is retained, the performance remains competitive but doesn't match the full model's efficacy. This suggests that while the Local view captures fine-grained interactions, the Global view's broader context is indispensable for achieving peak performance. Overall, the table illustrates the collective and individual significance of each component in the NMCLK model, emphasizing the need for a holistic approach in recommendation system design.

| Model | Contrastive loss | | Representation uniformity | | MovieLens-100k | | | | MovieLens-1M | | | |
|---|---|---|---|---|---|---|---|---|---|---|---|---|
| | Local | Global | Alignment | Uniformity | NDCG@20 | Hit@20 | MRR@20 | Recall@20 | NDCG@20 | Hit@20 | MRR@20 | Recall@20 |
| **NMCLK** | Y | Y | Y | Y | **0.2790** | **0.8780** | **0.4156** | **0.3528** | **0.2313** | **0.8598** | **0.3958** | **0.2484** |
| NMCLK without added Noise | Y | Y | Y | Y | 0.2709 | 0.8685 | 0.4006 | 0.3427 | 0.2219 | 0.8538 | 0.3800 | 0.2401 |
| NMCLK without CL | N | N | Y | Y | 0.2720 | 0.8759 | 0.4078 | 0.3473 | 0.2251 | 0.8425 | 0.3890 | 0.2348 |
| NMCLK without alignment | Y | Y | N | N | 0.2752 | 0.8738 | 0.4072 | 0.3508 | 0.2286 | 0.8566 | 0.3897 | 0.2461 |
| NMCLK without CL without alignment | N | N | N | N | 0.2715 | 0.8749 | 0.4026 | 0.3436 | 0.2211 | 0.8493 | 0.3924 | 0.2403 |
| NMCLK without Global view | N | N | N | N | 0.2581 | 0.8515 | 0.3868 | 0.3273 | 0.2067 | 0.8328 | 0.3574 | 0.2240 |
| NMCLK without Global view | Y | Y | N | N | 0.2708 | 0.8664 | 0.4126 | 0.3403 | 0.2256 | 0.8526 | 0.3856 | 0.2421 |
| NMCLK without Global view | N | N | Y | Y | 0.2720 | 0.8749 | 0.4066 | 0.3432 | 0.2257 | 0.8425 | 0.3901 | 0.2356 |
| NMCLK without Global view | Y | Y | Y | Y | 0.2725 | 0.8770 | 0.4159 | 0.3446 | 0.2218 | 0.8397 | 0.3876 | 0.2320 |

Table 2: Individual Component Analysis of NMCLK model

## 1.3 GENERALIZABILITY OF FRAMEWORK IN CTR TASKS:

To evaluate the generalizability of our framework in Click-Through Rate (CTR) prediction tasks, we conduct a compatibility study by incorporating NMCLK into various CTR prediction models, such as FM, FwFM, DeepFM, AutoInt, DCN, and DCNV2, as shown in Table 3.

The results show that our framework, when combined with various CTR models, consistently outperforms the corresponding base models across different datasets, as evidenced by the improvement in AUC and Logloss metrics. For instance, the combination of NMCLK and DCN achieves an AUC of 0.8147 and a Logloss of 0.5325 on the MovieLens-1M dataset, compared to the AUC of 0.8134 and Logloss of 0.5348 obtained by DCN alone. Similarly, the combination of NMCLK and AutoInt yields an AUC of 0.8032 and a Logloss of 0.5522 on the Ta-feng dataset, compared to the AUC of 0.7997 and Logloss of 0.5713 obtained by AutoInt alone.

These results demonstrate the effectiveness and generalizability of our framework in improving the performance of various CTR prediction models on different datasets.

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

| Model | MovieLens-1M | | Gowalla | | Ta-feng | |
|---|---|---|---|---|---|---|
| | AUC | Logloss | AUC | Logloss | AUC | Logloss |
| FM Rendle (2010) | 0.772 | 0.7409 | 0.5812 | 2.017 | 0.6306 | 2.701 |
| NMCLK$_{FM}$ | **0.7733** | **0.733** | **0.5842** | **1.894** | **0.6321** | **2.38** |
| FwFM Pan et al. (2018) | 0.7763 | 0.8453 | 0.5797 | 2.145 | 0.6277 | 3.411 |
| NMCLK$_{FwFM}$ | **0.7768** | **0.8317** | 0.5758 | **2.021** | **0.6295** | **2.875** |
| DeepFM Guo et al. (2017) | 0.7882 | 0.6479 | 0.6034 | 1.388 | 0.6883 | 0.9194 |
| NMCLK$_{DeepFM}$ | **0.7884** | **0.6443** | **0.6041** | **1.356** | **0.6898** | **0.8636** |
| AutoInt Song et al. (2019) | 0.7997 | 0.5713 | 0.6086 | 1.931 | 0.6805 | 1.46 |
| NMCLK$_{AutoInt}$ | **0.8032** | **0.5522** | 0.6017 | **1.424** | **0.682** | **1.413** |
| DCN Wang et al. (2017) | 0.8134 | 0.5348 | 0.6214 | 1.341 | 0.7007 | 0.8943 |
| NMCLK$_{DCN}$ | **0.8147** | **0.5325** | **0.6285** | **1.281** | **0.7017** | **0.8899** |
| DCNV2 Wang et al. (2021) | 0.806 | 0.5483 | 0.6071 | 1.359 | 0.6799 | 1.043 |
| NMCLK$_{DCNV2}$ | **0.8053** | **0.5466** | **0.6099** | **1.35** | **0.6834** | **0.9858** |

Table 3: Compatibility study in CTR prediction tasks

Canton of Geneva, CHE, 2018. International World Wide Web Conferences Steering Committee. ISBN 9781450356398. doi: 10.1145/3178876.3186040. URL https://doi.org/10.1145/3178876.3186040.

Steffen Rendle. Factorization machines. In *2010 IEEE International Conference on Data Mining*, pp. 995–1000, 2010. doi: 10.1109/ICDM.2010.127.

Weiping Song, Chence Shi, Zhiping Xiao, Zhijian Duan, Yewen Xu, Ming Zhang, and Jian Tang. Autoint: Automatic feature interaction learning via self-attentive neural networks. In *Proceedings of the 28th ACM International Conference on Information and Knowledge Management*, CIKM '19, pp. 1161–1170, New York, NY, USA, 2019. Association for Computing Machinery. ISBN 9781450369763. doi: 10.1145/3357384.3357925. URL https://doi.org/10.1145/3357384.3357925.

Ruoxi Wang, Bin Fu, Gang Fu, and Mingliang Wang. Deep amp; cross network for ad click predictions. In *Proceedings of the ADKDD'17*, ADKDD'17, New York, NY, USA, 2017. Association for Computing Machinery. ISBN 9781450351942. doi: 10.1145/3124749.3124754. URL https://doi.org/10.1145/3124749.3124754.

Ruoxi Wang, Rakesh Shivanna, Derek Cheng, Sagar Jain, Dong Lin, Lichan Hong, and Ed Chi. Dcn v2: Improved deep amp; cross network and practical lessons for web-scale learning to rank systems. In *Proceedings of the Web Conference 2021*, WWW '21, pp. 1785–1797, New York, NY, USA, 2021. Association for Computing Machinery. ISBN 9781450383127. doi: 10.1145/3442381.3450078. URL https://doi.org/10.1145/3442381.3450078.