# OpenReview forum: "NOISY MULTI-VIEW CONTRASTIVE LEARNING FRAMEWORK FOR ENHANCING TOP-K RECOMMENDATION"
_ICLR.cc/2024/Conference — Submitted to ICLR 2024_

### Official Review · Reviewer_g3vX · 2023-10-28

**Soundness:** 3 good
**Presentation:** 3 good
**Contribution:** 1 poor
**Rating:** 3
**Confidence:** 4

**Summary:**

To improve the data sparsity issue, this paper proposes a multi-view contrastive learning framework for knowledge-aware recommendation. The proposed model generates multiple modeling views including a collaborative view, a semantic view, and a structural view. The three views are constructed by utilizing part of the heterogeneous data combining the user-item interaction information and the item-entity knowledge information. Then two contrastive learning losses, and an alignment and uniformity loss is applied for supervision enhancement. Experiments on ML-100K and ML-1M validate the effectiveness of the proposed approach.

**Strengths:**

- Clear presentation. The paper is well-writen with good illustration figures. The introduction clearly highlights the major research motivation and key contributions of this paper. And it has a clear structure to introduce the view generation part and the contrastive learning part, respectively. I find it easy to follow.
- Important research topic. The paper targets an important research topic, namely self-supervised learning for knowledge-aware recommender systems.
- Technical design. The contrastive learning method adopted in this paper is conducted from multiple dimensions, including the cross-view contrastive learning, and the alignment and uniformity constraints.

**Weaknesses:**

- Limited novelty. There have already been some self-supervised learning approaches proposed for knowledge-aware recommendation (e.g. KGCL [1], KGIC [2], KACL [3]). In terms of view generation and contrastive constraints, this paper does not make sufficiently innovative contribution to this topic compared to the existing works.
- Insufficient experiments. i) The empirical study is conducted on two small-size datasets that are not aligned with the recent studies on knowledge-aware recommendation. ii) The paper does not involve the existing contrastive KG recommendation methods as baselines. iii) There is only the overall performance comparison. There lack other experiments such as ablation study, hyperparameter study, anti-noise investigation, for a comprehensive empirical study.
- Important part of methodology is not clearly explained. I cannot find the specific definitions for some of the self-supervised learning loss terms, such as $\mathcal{L}_u^g$, $\mathcal{L}_i^g$, $\mathcal{L}_u^l$, $\mathcal{L}_i^l$, and $L_{local}$. Please correct me if I am wrong.

Minor mistakes:
- In the task formulation section, there should be braces in the definitions for user/item sets, the knowledge graph, and so on.
- LightGCN is the original model name. Adding a hyphen in the name (Light-GCN) may cause confusion. If the used GCN architecture is not exactly LightGCN, using other expressions like light-weight GCN would be better.
- The paper utilizes $\mathcal{L}$ for loss terms in most cases, but sometimes $L$ is used. The notations could be better if unified.
- Typo: "This results ..." in page 5

[1] Knowledge Graph Contrastive Learning for Recommendation

[2] Improving Knowledge-aware Recommendation with Multi-level Interactive Contrastive Learning

[3] Knowledge-Adaptive Contrastive Learning for Recommendation

**Questions:**

- What are the major improvements brought by the proposed method, in comparison to the existing CL methods for KG recommendation listed above?
- How to calculate the CL loss terms in detail?

---

### Official Review · Reviewer_1tBK · 2023-10-31

**Soundness:** 2 fair
**Presentation:** 3 good
**Contribution:** 1 poor
**Rating:** 5
**Confidence:** 3

**Summary:**

This manuscript proposes to use contrastive learning for enhancing top-k recommendation. Alignment and uniformity constraint module are introduced to both the global and local parts. Experimental results on two datasets show that the proposed NMCLK outperforms previous methods.

**Strengths:**

This paper is written clearly and easily understandable.
The experimental results seem to be good on two datasets.

**Weaknesses:**

The main contribution of this manuscript is introducing self-supervised learning methods which are commonly used in CV and NLP to the recommendation field. I think the novelty is not enough for an ICLR paper.
The contrastive learning is to make features from the same field similar and in contrast, with large distances for disparate ones. Is that always true in recommendation?
The ablation studies are not convincing, the authors should conduct experiments with and without each proposed component.

**Questions:**

See the weakness part.

---

### Official Review · Reviewer_8SVs · 2023-10-31

**Soundness:** 2 fair
**Presentation:** 2 fair
**Contribution:** 1 poor
**Rating:** 3
**Confidence:** 3

**Summary:**

This paper presents a novel Noisy Multi-view Contrastive Learning framework for Knowledge-aware recommender systems (NMCLK). NMCLK generates three different views over user-item interactions and knowledge graphs, and further introduces a noise addition module to improve model robustness. The three views include a global-level structural view, a local-level user-item view, and an item-item semantic view. NMCLK also utilizes representation loss and uniformity loss to enhance the quality of the learned user-item representations. Experimental results on two movie recommendation datasets demonstrate that the proposed method outperforms state-of-the-art approaches.

**Strengths:**

- NMCLK is able to outperform a number of baseline approaches on two movie recommendation datasets.

**Weaknesses:**

- The novelty of the paper is limited. The authors should derive more insights in the alignment and uniformity constraints in contrastive learning, and maybe design a better contrastive module.
- The design choice of the contrastive module is not explained. In addition, the readers cannot know how and why the multi-view framework works, and cannot see the performance improvement after using the multi-view framework.
- The model is only evaluated on NMCLK on two movie recommendation datasets, and it is questionable whether the performance of NMCLK will generalize to other recommendation domains.
- No ablation study is conducted to verify the effectiveness of the introduced modules, e.g., the noise module and the contrastive learning module.

**Questions:**

- wu2022noisytune and simgcl are not properly cited.
- Section 5.1.1 says "Table 1 displays the statistics of the three datasets mentioned above", while only two datasets are used.

---

### Official Review · Reviewer_X5Ei · 2023-11-08

**Soundness:** 2 fair
**Presentation:** 3 good
**Contribution:** 2 fair
**Rating:** 5
**Confidence:** 3

**Summary:**

This paper aims to utilize multi-view contrastive learning to enhance the recommender system. The proposed model based on self-supervised learning can aggregate the information from item knowledge graph, item similarity and historic records. Extensive experiments verify the effectiveness of the proposed model.

**Strengths:**

1.	The writing quality of this paper is high.
2.	This paper introduces a novel contrastive learning model to enhance the CTR recommendation task.

**Weaknesses:**

1.	This paper lacks theoretical analysis to clarify how each contrastive learning module benefits the final CTR task.
2.	It would be beneficial to test the proposed model in various datasets across different domains, not just in the movie domain.
3.	A comprehensive ablation study is necessary to clarify the effectiveness of each proposed module, including the contrastive module and the feature alignment module.
4.	There is a citation format error below Equation (2): "Inspired by SimGCL (simgcl) and Noisytune wu2022noisytune papers' additions."

**Questions:**

See weaknesses.

---

### Meta-Review · Area_Chair_ALVH · 2023-12-27

**Metareview:**

The authors propose a multi-view contrastive learning method for knowledge-aware recommendation systems, referred to as the "Noisy Multi-view Contrastive Learning Framework (NMCLK), to improve top-k recommendations. Specifically, NMCLK generates three different views over user-item interactions and item-entity knowledge (i.e., knowledge graphs) to capture a global-level structural view, a local-level collaborative view, and an item-item semantic view. Using a specific encoding method for each view, the authors incorporate recent advances regarding self-supervised learning to refine feature representations using global-level contrastive learning and alignment and uniformity constraints to derive more discriminative representations. Experiments are conducted on MovieLens-100k and MovieLens-1M against strong baseline recommender systems and demonstrating strong performance on widely used metrics (e.g., NDCG@k, HIT@k, MRR@k, Recall@K).

Consensus reviewer strengths regarding this submission include:
- The proposed method is clearly presented and the paper is easy to understand. Illustrations add to the understanding and help motivate the concepts that drive this work.
- (Multi-view) Contrastive learning has been shown successful in other domains, but there has been limited adaption to the important RecSys domain (to the best knowledge of the reviewers). This submission did this sensibly and successfully (and empirically better than existing work).
- The empirical performance is strong relative to a number of recent relevant systems on widely-used benchmark datasets.

Conversely, consensus limitations included:
- Limited novelty: This work is essentially an adaptation of existing methods from related ML domains to RecSys. While done successfully, I believe the authors were looking for a clearer description of the specific work that this draws from and the precise methodological contributions. Additionally, there should be a more explicit contrast with the most RecSys recent work that does use self-supervised learning in the discussion of contributions. Ideally, there may even be possibly application-specific modifications beyond a 'direct transfer' from existing work.
- Insufficient experiments: There are three dimensions in this regard. First of all, experiments are only performed on MovieLens; the KGIN paper alone uses Amazon-Book, LastFM, and Alibaba-iFashion. While these may not all be necessary, only using one domain limits evidence for this working more generally (and these all have the material to perform in knowledge-aware settings). Second, the reviewers believe that an ablation study is needed to tease apart the relative contributions of the various methodological contributions; the reported results are primarily only'top-line' experiments. Third, additionally experiments regarding sensitivity analysis of hyperparameters, etc. would further strengthen the paper.
- From my own reading, it isn't clear that this is a new SotA when comparing to recent methods that incorporate LLMs, etc. as knowledge sources/embeddings. While the contribution may be sufficient without this direct comparison, there should be some discussion regarding this if it is a limitation to further add precision to the contributions of this work.

Overall, the reviewers seems to believe that this is a good step in deriving contrastive representations for RecSys with promising empirical results. However, a clearer specification of the methodological contributions and additionally empirical evaluations are needed to support the claims made in the paper (and clearly establish that this is a SotA method) before the paper is ready for publication in a top-tier venue.

**Justification For Why Not Higher Score:**

The reviewers raised consistent and valid concerns regarding novelty of the method, better contextualization with respect to existing work to clearly state the contributions, and more empirical validation.

**Justification For Why Not Lower Score:**

N/A

---

### Decision · Program_Chairs · 2024-01-16

Reject